# The Orexin System and Its Impact on the Autonomic Nervous and Cardiometabolic System in Post-Acute Sequelae of COVID-19

**DOI:** 10.3390/biomedicines13030545

**Published:** 2025-02-21

**Authors:** Jana Ruhrländer, Styliani Syntila, Elisabeth Schieffer, Bernhard Schieffer

**Affiliations:** 1Department of Cardiology, Angiology and Critical Care Medicine, Philipps University Marburg, 35043 Marburg, Germany; jana.ruhrlaender@uk-gm.de (J.R.); syntila@med.uni-marburg.de (S.S.); elisabeth.schieffer@uni-marburg.de (E.S.); 2State of Hessen Post-COVID Coordination Center, 35043 Marburg, Germany

**Keywords:** orexin, COVID-19, PASC, inflammation, cardiovascular system, autonomic nervous system

## Abstract

Orexins (OXs) are critical for regulating circadian rhythms, arousal, appetite, energy metabolism, and electrolyte balance, affecting both the autonomic nervous system (ANS) and the cardiovascular system (CVS). Disruption of the OX system can result in symptoms similar to those observed in post-acute sequelae of COVID-19 (PASC). This review emphasizes the adverse effects of OX dysregulation on autonomic and cardiometabolic functions in patients with PASC. Additionally, we highlight the potential of anti-OX therapies to provide neuroprotective, anti-inflammatory, and immunoregulatory benefits, offering hope for alleviating some of the debilitating symptoms associated with PASC.

## 1. Introduction

In 1998, a pair of neuropeptides was first described by two independent groups [1,2]. Due to their expression in neurons of the dorsolateral hypothalamus and structural similarity to the peptide family of incretins, they were initially referred to as hypocretins. The name orexins (OXs) is derived from the Greek word *orexis*, which means appetite, as intraventricular administration of both peptides promotes food intake dose-dependently [3]. They are formed from the polypeptide prepro-orexin (prepro-OX), which is proteolytically cleaved into the active peptides orexin-A and orexin-B (OX-A and OX-B). Of these, only OX-A binds to the orexin receptor 1 (OX_1_R), while both bind to orexin receptor 2 (OX_2_R) [4]. Following a nomenclature proposal by the International Union of Basic and Clinical Pharmacology, the gene and the mRNA are now mainly referred to as hypocretin neuropeptide precursor *(HCRT)* and the precursor peptide and the processed peptides as orexins. According to the same principle, the two G-protein-coupled receptors (GPCRs) for the peptides are named OX_1_R and OX_2_R, the genes encoding them being *HCRTR1* and *HCRTR2* [3], via which the OXs influence calcium influx and adenyl cyclase activity [4].

The OX network interacts with other neurotransmitter systems, forming a complex network that tunes and maintains a variety of bodily functions. Disruptions to this system—whether due to hyposecretion, hypersecretion, loss of OX neurons, neuronal inflammation, or the presence of antibodies against OX receptors (OXR)—can lead to significant physiological impairments and debilitating symptoms.

## 2. Post-Acute Sequelae of COVID-19

According to WHO estimates, the COVID-19 pandemic and its aftermath may have claimed the lives of nearly 15 million people in 2020 and 2021 alone [5]. Additionally, an estimated 400 million people worldwide suffer from PASC, with an annual economic impact projected to reach one trillion US dollars [6]. These PASC disorders are often severe and carry a high risk of becoming chronic, leading to a significant decline in quality of life and imposing substantial socioeconomic consequences.

Over 200 possible symptoms of PASC have been identified, with fatigue, breathing problems, and cognitive impairment being the most prominent [7].

Here we will highlight the significant symptomatic overlap between disorders of the OX system and chronic post-viral conditions, such as PASC, examine the existing connections, and explore how OX-targeted therapies could offer novel treatment options for these conditions.

## 3. Distribution of Orexin Neurons and Receptors

Research on the distribution and projections of OX neurons, as well as the distribution of OXR has primarily been conducted in rodents. As a result, there is limited data available regarding humans. Comparisons of the distribution of OX neurons and their projections among different rodent species indicate that the OX system is highly conserved across these species [8]. Therefore, and due to the same functionality of some brain areas, it can be assumed that the distribution in humans is similar (Figure 1).

OX neurons project to almost the entire brain, in particular to cholinergic and monoaminergic nuclei of the brainstem, such as the locus coeruleus, which is responsible for attention, learning and sleep control; the raphe nuclei, which play an important role in pain perception; as well as the rostral ventrolateral medulla (RVLM), which controls sympathetic activity and thus also cardiovascular functionality. In addition, they are present in various areas of the hypothalamus that are responsible for the control of sleep, attention, eating behavior, and the control of autonomic functions, such as the tuberomammillary nucleus, the ventrolateral preoptic nucleus, the arcuate nucleus, the hypothalamic paraventricular nucleus (PVN), as well as in the ventral tegmental area (VTA) in the midbrain, which plays a central role in the endogenous reward system [9,10].

OX_1_R and OX_2_R exhibit distinct expression patterns, with OX_1_R prominent in the ventromedial hypothalamic nucleus, while OX_2_R is predominantly found in the PVN [11]. In the human hypothalamus, OX_1_R was found in the lateral hypothalamic area, the lateral preoptic nucleus, the supraoptic nucleus, the dorsomedial nucleus, the ventromedial nucleus, and the PVN, along with a meager number in the mammillary bodies [12].

A study on the distribution of OX_2_R in the peripheral tissue of male rats revealed only very low OX_2_R protein expression. While it was highest in the adrenal cortex and the kidney cortex, only a negligible density was found in the heart, testes, adrenal medulla, intestinal mucosa, intestinal muscle, pituitary gland, skeletal muscle, thyroid gland, and lung and bladder muscle [13]. The density of OX_2_R protein expression differed from the analyzed amount of OX_2_R mRNA in male and female rats. Here, a significant amount of OX_2_R mRNA was found in the adrenal glands of male rats, which was not reflected in the amount of expressed OX_2_R, and a large amount of prepro-OX mRNA was found in the testes of male rats. In addition to the brain, OX_1_R mRNA was found in the pituitary gland and only small amounts in the kidney, adrenal glands, thyroid, testis, ovaries, and jejunum [14].

Although only low densities of OX_2_R and low amounts of OX_1_R mRNA and OX_2_R mRNA were found in the peripheral tissue, this, as well as the finding of prepro-OX mRNA, indicates an effect of the OX system in the periphery.

The presence of OXs and OXRs is widespread in the human gastrointestinal tract. OX-A-immunoreactive fibers and OXR-expressing cells or cells with OXR-like immunoreactivity were found in the stomach and the entire human intestine and pancreas [15].

While an older study found no prepro-OX and OXR mRNA in the muscle and other areas of the body of male rats [14], a later study found the presence of OX_2_R in several of these areas, including skeletal muscle, albeit in negligible amounts [13].

## 4. Orexins Affect the Cardiometabolic System

These OX-containing fibers innervate several cerebral areas important in the regulation of cardiovascular function. These projections into important brain areas for cardiovascular functionality, such as the RVLM, but also the low occurrence of OX_2_R directly in the heart, indicate their importance for the CVS. In this regard, Chen et al. demonstrated that OX injections in anesthetized rats led to a dose-dependent increase in mean arterial pressure (MAP) and heart rate (HR) [16]. In conscious rats, OX administration enhances MAP, HR, and renal sympathetic nerve activity; higher doses of OXs increase plasma catecholamine levels, suggesting that OXs contribute to sympathetic nerve activity [17].

It is therefore not surprising that OXs are also involved in cardiovascular pathophysiological processes, e.g., in the regulation of stress-induced hypertension by regulating the nitric oxide synthase/nitric oxide (NOS/NO) system. Studies on the cardiovascular effects of NO and the soluble guanylyl cyclase/cyclic guanosine monophosphate (sGC/cGMP) signaling cascade have so far come to different conclusions. Presumably, NO can mediate both a vasopressor and a vasodepressant effect possibly depending on origin, locale, and dose differences [18].

Another study supports the hypothesis of sGC-mediated cardiovascular effects of NO and a vasopressor effect of neuronal NOS (nNOS)-derived NO, regulated by OX-A, in rats with stress-induced hypertension [19]. In contrast, dose-dependent effects were observed with microinjections of OX-A and OX-B into the nucleus tractus solitarii (NTS). A cardiovascular excitatory response by OXs was observed at higher doses, while at lower doses they produced the opposite effect. In this case, the OXs mediated a cardiovascular depressant effect via nNOS-derived NO and the sGC pathway, and could accordingly be reduced by an NOS, an nNOS, and an sGC inhibitor, but not by an inducible NOS (iNOS) or eNOS inhibitor. The vasopressor effect of high-dose OX could be canceled by the simultaneous administration of a γ-aminobutyric acid A (GABA_A_), and a GABA_B_ receptor antagonist and could also be partially attenuated by an NOS inhibitor in a dose-dependent manner [20].

The dose- and site-dependent effects are partly different, and the OX-A and OX-B injections can have a sympatho-excitatory effect, but a complete loss of OX-A and OX-B in OX knockout mice apparently leads to a reduced lower sympathetic vasoconstrictor tone [21] as well as co-localization of OX_1_R and nNOS and iNOS cells in the RVLM [19]. This shows how complex the interplay is between OXs, the NO/sGC cascade and other neurotransmitter systems, and how further research is needed in this area.

Intranasal OX-A administration has shown mixed effects on sympathetic baroreflex function in humans. While one study found no significant changes in BP, HR, or muscle sympathetic nerve activity (MSNA) after OX-A administration [22], another later study from the nearly same team found that although intranasal OX-A application has no direct effect on BP, HR, heart rate variability (HRV), and baroreflex sensitivity, it increases resting vasoconstrictive MSNA, suggesting a potential role in resetting the vascular baroreflex threshold upward [23] and thereby indirectly regulating the physiological baroreflex through its role in sympatho-adrenergic-induced vasoconstriction. Furthermore, the involvement of OX in sympathetic nerve regulation was shown in an orexin/ataxin-3 transgenic rat model. The rats with a lower number of OX neurons showed decreased baroreflex and β-adrenergic blocker responsiveness than those with a physiological number of OX neurons [24]. Another team also found a resetting of the baroreflex in both prepro-OX knockout mice and orexin/ataxin-3 transgenic mice, in addition to lower basal arterial blood pressure and an attenuated stress response [25]. Although intranasal administration of OX does not directly affect BP, HR, and baroreflex sensitivity, the new studies that found increased MSNA suggest that OX does play a higher-level role in the regulation of the baroreflex, which is supported by the studies in prepro-OX knockout mice and orexin/ataxin-3 transgenic mice and rats.

Postural orthostatic tachycardia syndrome (PoTS) is a hallmark of PASC characterized by autonomic dysfunction and impaired adaptation to orthostasis. PoTS is further characterized by pathologic tachycardia in an upright position with symptoms like palpitations, anxiety, and shortness of breath as indicators of sympathetic activation [26]. PoTS occurs in a large number of patients with PASC [27,28].

It was shown that PoTS patients exhibit exaggerated MSNA responses due to activations of the baroreflex system compared to healthy controls [29]. However, tachycardia during orthostasis and exercise in PoTS patients is not attributed to abnormal baroreflex control of HR when corrected for differences in HR [30]. An excess of catecholamines (norepinephrine) is also found in PoTS patients suffering from the hyperadrenergic form of the disease [31]. These findings suggest that alterations in autonomic reflexes play a crucial role in the pathophysiology of PoTS, although the exact mechanisms remain complex and multifaceted.

However, the exact mechanisms by which OXs induce cardiovascular reactions are not yet fully understood. Their effects on the cardiovascular system are likely mediated through the activation of calcium/calmodulin-dependent kinase II (CaMKII) expression within the PVN and key cardiovascular control regions [32]. Besides the influence on cardiovascular function, the PVN plays a key role in autonomic regulation [33] and controls pituitary–adrenocortical activity, stress response, fluid homeostasis, analgesia, food intake, and gastrointestinal functions [34].

OX dysfunction has already been discussed as an explanation for the pathophysiology of stress-induced heart disease, Takotsubo cardiomyopathy [35], which has been occurring increasingly since the COVID-19 pandemic, both after COVID-19 infections as well as in the general population, and after COVID-19 vaccinations [36,37,38]. Takotsubo is related to excess catecholamine and can occur in cases of extreme autonomic response [35].

OXs also influence the CVS on the endothelial level. OX-A can suppress endothelial cells inflamed by oxidized low-density lipoprotein (ox-LDL) through the partial inhibition of the ox-LDL mediated adhesion of monocytes to endothelial cells, which is achieved by reducing MAPK p38 and NF κB activation via its receptor OX_1_R [39].

Endothelial dysfunction is a known problem in both acute COVID-19 Infection [40] and in myalgic encephalomyelitis/chronic fatigue syndrome (ME/CFS), the most severe form of PASC [41].

Vascular endothelial dysfunction can be a consequence of chronic high glucose exposure, among other things. OX-A exerts a strong protective effect against glucose-induced activation of the NLR family pyrin domain containing 3 (NLRP3) inflammasome in vascular endothelial cells [42]. Furthermore, an angiogenic activity could be demonstrated, in particular for OX-A, which is mediated via the MEK/ERK signaling pathway in endothelial cells. In a Matrigel plug assay in mice, neovascularization could be induced in vivo by OX-A and ex vivo sprouting of endothelial cells in rat aortic rings. The OXRs also increase the migration and tube formation of human umbilical vein endothelial cells (HUVECs) and activate the extracellular signal-regulated kinase 1/2 (ERK1/2) in HUVECs [43]. The influence of OXs on energy homeostasis is largely due to their general importance in weight, food intake, and overall metabolism.

In human HEK293 cells expressing OX_1_R, OX-A was shown to upregulate 260 genes at least twofold, while 64 were found to be downregulated by twofold or more. More than half of the upregulated genes are important either for cell growth or metabolism. The downregulated genes are largely uncharacterized; the second largest part is related to metabolism, followed by cell–cell communication, which underscores their regulatory impact on metabolic pathways [44].

In vitro, OXs not only promote the release of norepinephrine but also enhance epinephrine release from cultured bovine adrenal medullary cells [45]. Based on these findings, and the observation that OX administration in various brain regions increases MAP, HR, and sympathetic nerve activity, it is tempting to speculate that OXs control a multilevel, multistep blood-pressure regulation mechanism that involves peripheral (endothelial cells), organ (adrenal medullary cells), and central (brain, NTS) regulatory mechanisms.

In general, many research results indicate the significant role of OXs in glucose, insulin, and lipid metabolism [46,47]. They affect glucose uptake into adipocytes and triglyceride accumulation [47], lipolysis [48], and the development and function of brown adipose tissue (BAT) [49]. An imbalance in the OX system can therefore lead to a variety of metabolic disorders such as dyslipidemia, as shown by studies that linked OX gene expression with hypertriglyceridemia [50] and insulin resistance [51], and could be important for the development of type 2 diabetes [46], for which, as for insulin resistance, there is an increased risk following COVID-19 infection [52]. Also, increased risk and higher burden of dyslipidemia have been associated with COVID-19 [53], as well as the impact of statin therapy on lipid metabolism and therefore on COVID-19 disease [54]. The diabetes drug metformin showed significant protection against the development of PASC in a multicenter, randomized, quadruple-blind, parallel-group, phase 3 trial [55].

OXs regulate glucose homeostasis on the one hand by stimulating active glucose uptake mediated by phosphoinositide 3-kinase (PI3K)- via the glucose transporter type 4 (GLUT4), and by enhancing insulin-induced glucose uptake [47].

The influence on lipid metabolism can be explained on the one hand by the inhibition of lipolysis, but also by increased peroxisome proliferator-activated receptor-γ (PPARγ) expression, which could be shown in vitro and which increased lipogenesis. Furthermore, OX-A influences glucose and lipid metabolism by increasing plasma adiponectin [47].

It has been shown that hypoglycemia activates the OX system [56] and that OX-A can increase blood glucose and insulin levels, while OX-B only increases insulin levels [57].

Due to the close connection between OXs and food intake, it is not surprising that they are also important for the development of obesity and associated hypertension [58]. However, the exact effect of OXs on body weight is complex. Yamanaka et al. showed that short-term chronic administration of OX-A more than doubled the food intake of rats during the daytime, but weight, glucose levels, total cholesterol, and free fatty acids remained the same [59]. In turn, a study in mice showed that loss of OX neurons leads to late-onset obesity despite reduced food intake [60], while overexpression of OXs leads to mice becoming resistant to weight gain caused by increased calorie intake, as do OX deficient mice [61].

## 5. The Neuronal Orexin Neurotransmitter System and Neurocognitive Dysfunction and Neuroinflammation Associated with Post-Acute Sequelae of COVID-19

OX neurons found in the lateral hypothalamus (LH) and perifornical area (PFA) are in exchange with many neurotransmitter systems, and form negative and positive feedback loops, for example with serotonin, noradrenaline, dopamine (DA), histamine, acetylcholine, and Neuropeptide Y (NPY) [10], but also GABA, corticotrophin-releasing factor (CRF), neurotensin (NT), thyrotropin-releasing hormone (TRH), melanin-concentrating hormone (MCH) [62], glutamate (Glut) [63], and endocannabinoids (eCB) [64] (Figure 2).

They also contribute to the modulation of the network and their activity by releasing the excitatory neurotransmitters OX and Glut, as well as inhibitory neurotransmitters such as dynorphin (Dyn), and nociceptin/orphanin FQ (N/OFQ). This self-regulation of the OX neurons takes place through the time-dependent release of more excitatory (OX, Glut) or inhibitory (Dyn, N/OFQ) neurotransmitters [62]. Although the excitatory effect ultimately predominates, it can be temporarily masked by the inhibitory effect of Dyn and N/OFQ. Even on MCH neurons, which are affected by the same effect, an overlapping of the excitatory effect is likely over a longer period, while MCH can indirectly inhibit excitatory inputs through presynaptic Glut release [62].

While serotonin and noradrenaline exert an inhibitory effect on OX neurons, the effect of DA can be contrasted via the activation of different receptors (D1 and D2 receptors) [65].

CRF and NT co-expressed in CRF neurons have an excitatory effect on OX neurons, as does TRH, which depolarizes OX neurons. However, it also increases the activity of presynaptic GABAergic interneurons, which is why it could also have an indirect inhibitory effect [62].

Via retrograde transmission, eCB provides an inhibitory effect on OX neurons, while a cannabinoid agonist has been shown to depolarize MCH neurons and increase their firing rate [64].

There are links between the OX system and all of the neuro-psychiatric subgroups defined by Davis et al., namely cognitive dysfunction, speech and language, memory, headaches, smell and taste, sleep, emotion and mood, hallucinations, and sensorimotor [7]. Some of the connections are well known and studied, such as sleep [66,67] and emotion and mood [10]. OXs also contribute to functioning memory and cognition, promote neurogenesis in the hippocampus, and improve spatial learning [10]. Conversely, OX deficiency or dysfunction has been linked to cognitive impairments and learning and memory deficits [10,68], which all also influence speech and language. They are linked to pain regulation in general, and also primary headaches and medication overuse headaches [69,70]. OX neurons project into the entire olfactory pathway and increase olfactory sensitivity [71,72]. It also appears that OXs may play a role in modulating taste preferences and enhancing hedonic responses to sweet flavors [73,74]. Hallucinations and sleep paralysis are symptoms that occur in narcolepsy, a disease that is closely linked to the OX system and loss of OX neurons and disturbances in it [75]. The fact that hallucinations can also occur, although rarely, as side effects with OXR antagonists shows that there is probably a connection here [76,77]. OXR antagonists may also be effective in the treatment of sensory and motor symptoms in conditions such as restless leg syndrome [78], and their association with inflammatory neuropathies [79] also suggests that the OX system plays a role in the modulation of sensory and motor processes.

The significance of disorders in the OX system for symptoms affecting the nervous system in PASC and similar diseases therefore seems very likely, even if it cannot be said whether they are the cause or only a consequence of other processes.

Neuroinflammation is a process that plays a role in many neurological diseases such as Alzheimer’s disease and Parkinson’s disease [80], and could also be a cause of the neurocognitive symptoms of PASC. Although there have been no broad-based studies on neuroinflammation in the context of PASC, a connection between neurocognitive symptoms and inflammatory processes in the brain in PASC can be assumed [81,82]. There is evidence for neuroinflammatory processes in acute COVID-19 infections [83] and after a Covid infection, the risk of cerebrovascular events remains increased, including those that may be related to OX dysregulation such as ischemic and hemorrhagic strokes, cognitive and memory impairment, peripheral nervous system disorders, migraine, seizures, mental disorders, Guillain–Barré syndrome, encephalitis, or encephalopathy [84].

Neuroinflammation is a complex reaction of the brain to damage, infections, or imbalances and is triggered by activation of the immune system in the central nervous system (CNS), in particular microglia and astrocytes, but also mast cells [85]. Although microglia may also play a role in the restoration of injured brains [86], chronic activation of microglia is a key component of neuroinflammation and could therefore be a plausible cause of neuroinflammation in PASC. It has also been shown in a mouse model of ME/CFS [87], and indirectly via detecting a ligand for PET for a translocator protein expressed by activated microglia or astrocytes [88].

Microglia and various neurotransmitters appear to have a bidirectional connection [89]. Microglia express receptors for various neurotransmitters, and the activation of some of them—GABAergic, cholinergic, and adrenergic—inhibits the activity of microglial cells. ATP and adenosine have an activating effect, and Glut and DA can exert both an activating and an inhibiting effect depending on the receptor [90].

Conversely, the microglia cells influence neurotransmitters, such as Glut, and can thus lead to impaired synaptic signaling and cognitive deficits [89,91].

One possible therapeutic approach that has shown success in some patients with PASC is treatment with low-dose naltrexone [92], an opioid receptor antagonist that has an immunomodulatory effect via the modulation of Toll-like receptor 4 (TLR4), which signals and counteracts microglial activation [93].

Various studies have already demonstrated the neuroprotective effect, particularly for OX-A. It was shown that OX-A significantly improves the outcome of mice with ischemic brain injury by reducing inflammation through the modulation of Interleukin-6 (IL-6) and tumor necrosis factor α (TNF-α) in microglia [94].

In an experiment with intranasal OX application, the amount of inflammatory markers such as interleukin-1β (IL-1β), iNOS, TNF-α, glial fibrillary acidic protein, and microglial cell surface marker (CD11b) in the hypothalamus was significantly reduced, while OX_1_R and OX_2_R expression was significantly increased [95].

Immunological dysfunction with increased pro-inflammatory cytokines has also been described in PASC [96] including the p38 MAPK signaling pathway [97].

Together with the findings of Tsagkaris et al. on COVID-19-associated neuroinflammation by cytokine storms [98] and the increased risk of neurodegenerative diseases after COVID-19 infections, a treatment trial with intranasal OX could be an option for PASC patients with severe neurocognitive impairment.

Not only acute COVID-19 infections, but also COVID vaccines, can trigger various symptoms, called post-acute COVID-19 vaccination syndrome (PACVS), which may be due to the spike protein as the same triggering factor [99]. However, similar effects were observed with other vaccinations, suggesting a secondary hyperinflammation effect [100]. Interestingly, the cases of narcolepsy following Pandemrix H1N1 vaccination already showed a link between presumed autoimmune post-vaccination phenomena and the OX system. All Pandemrix-associated narcolepsy cases were type 1 narcolepsy, which is characterized by a hypocretin deficiency in the CNS [101]. Also after infection with SARS-CoV-2, new onset narcolepsy was reported in a nine-year-old girl, with excessive daytime sleepiness, cataplexy, nightmares and violent behaviors during nocturnal sleep, obesity, agitation, anxiety, and aggression. A lumbar puncture revealed pathologically low OX-A levels [102]. In another report, a course of narcolepsy type 2, with borderline OX levels in the cerebrospinal fluid, was halted by the administration of methylprednisolone [103].

Inflammatory neuropathies and immune-mediated neurological disorders, such as Guillain–Barré syndrome, chronic inflammatory demyelinating polyneuropathy, neuromyelitis optica spectrum disorders, and autoimmune encephalitis, are associated with reduced OX levels [79,104], which have also been described as new onset or worsening of an existing disease after a COVID-19 infection or vaccination [105,106,107,108,109,110,111].

However, non-specific and sometimes transient symptoms such as general weakness, fatigue, sleep disturbances, loss of appetite, headaches, myalgias, pain, hormonal and menstrual disorders, and depression, which have been described after COVID-19 infection [7,112], could also be accompanied or triggered by a disturbed OX system.

One study showed that some patients with acute COVID-19 infection developed autoantibodies against OX_2_R. These were negatively correlated with exceptionally low Glasgow coma scores [113].

Due to their general anti-inflammatory effect owing to the decrease of pro-inflammatory cytokines and possibly also indirect regulation via the ANS, OXs can influence autoimmune phenomena through cytokine storm and hyperinflammation triggered by COVID-19 infection or vaccination [114,115].

OXs and cytokines interact with each other in a complex manner, as already addressed in the outcome of mice with ischemic brain injury [94]. In mice in a lipopolysaccharide-induced septic shock model, central and peripheral administration of OX-A increased the survival rate by suppressing excessive cytokine release, in particular in those that are microglia-derived [116]. In gut–brain communication, OX-A plays a bidirectional role, potentially modulating immune responses and reducing pro-inflammatory cytokine production in intestinal inflammatory diseases, restoring the integrity of the intestinal epithelial barrier [117]. In a mouse model with chemically-induced colitis, OX-A showed even better efficacy for healing than the monoclonal antibody infliximab, by reducing circulating cytokines, myeloperoxidase activity, lipopolysaccharide-binding protein, and cytokine gene expression [118].

In fibroblast-like synoviocytes, OX-A was shown to decrease TNF-α-induced secretion of IL-1β, IL-6, and IL-8, as well as the production of reactive oxygen species (ROS) by inhibiting the NF-κB signaling pathway. Conversely, TNF-α reduced the expression of OX_1_R [119].

On the other hand, a study on Sprague Dawley rats showed that microinjections of OX-A into the PVN can increase pro-inflammatory cytokines such as IL-1-β, IL-6, and TNF-α [120].

Even if the evidence in favor of an anti-inflammatory effect generally prevails, the interaction between cytokines and OXs is therefore very complex, and different effects can develop due to the different reactions of cell types, the site of action, or different signaling pathways. An OX-mediated natural mechanism to protect against hyperinflammation or a shift in the balance in a pro-inflammatory direction of the immune system through OX dysregulation could therefore play a role in the development and/or maintenance of a hyperinflammatory state or cytokine storm.

Another very well-known problem of long-term consequences after COVID-19 infection, which has also been described after COVID-19 vaccinations and is closely related to the OX system, is sleep disturbances [7,121]. The role of OXs in the regulation of sleep–wake cycles and alertness has already been well studied [66,67]. People with narcolepsy exhibit orexin neuronal loss [122], and in mouse models with genetic ablation of orexin neurons, the mice developed a clinical picture similar to human narcolepsy [60]. Narcolepsy patients also exhibit neurocognitive deficits in certain areas [123] which is also observed in general sleep deprivation [124]. Gene expression changes upon OX_1_R activation were analyzed in a cell line from hypothalamic neurons of mice into which the human OX_1_R was cloned. Interestingly, the observed transcriptional effects are strikingly similar to those observed in in vivo microarray studies during sleep deprivation, highlighting the importance of OX signaling for wakefulness and sleep deprivation [125].

OXR antagonists/agonists could therefore be a potential treatment option for sleep-related problems in PASC and related diseases. In various studies, they achieved good results in increasing wakefulness after sleep onset, latency to sustained sleep, and sleep efficiency [126].

## 6. Discussion

Disturbances in the orexin (OX) system can lead to complex disorders that affect both neurocognitive and cardiometabolic functions. These dysregulations are frequently observed in patients following an acute COVID-19 infection (post-acute sequelae of COVID-19, PASC). We assume that several symptoms that occur after COVID-19 are due to an imbalance in the OX system. In particular, factors such as vascular tone, cardiac and endothelial function, lipid metabolism, and neuropeptide regulation are crucial to the pathophysiology of the OX system, as well as PASC.

Hypo- or hypersecretion, as well as a loss of OX neurons or autoantibodies against OXR can have various bi-directional consequences. A lack of OX could generally lead to a pro-inflammatory status. Chronic inflammation can cause a variety of symptoms, including fatigue, muscle pain and cognitive impairment. PASC patients exhibit several behavioral and physiological changes that occur in response to infection or an inflammatory response by the body to conserve energy, such as fatigue, anorexia, lethargy, sleep disturbances, and social isolation. These typical disease reactions are closely linked to the OX system.

OXs influence on vasoconstriction could lead to an incorrect adjustment of the vascular tone, which could cause symptoms of circulatory dysregulation, as is often the case with PASC. Together with the influence they have on catecholamines and the adjustment of the baroreflex, this could explain phenomena such as PoTS, orthostatic hypotension, or newly occurring hypertension after COVID-19 infections.

SARS-CoV-2 attacks the endothelium and the vascular system, which can trigger long-term endothelial dysfunction that could contribute to the pathomechanism of PASC. OX deficiency could exacerbate this situation, as OX-A has a strong protective effect on the endothelium, can suppress endothelial inflammation, and has an angiogenic effect.

OXs influence overall metabolism by controlling food intake, energy expenditure, glucose, and fat metabolism. Various metabolic disorders are known in acute COVID-19 infections, as well as in the post-acute consequences, for example insulin resistance, dyslipidemia, and diabetes. However, unspecific symptoms such as fatigue and PEM could also be caused by metabolic disorders. The control over sympathetic activity exerted by OX could also lead to reduced alertness and attention in the presence of dysregulation.

Specifically for sleep, OXs are essential, and newer therapeutic options for sleep disorders directly target the OX system. PASC is characterized by a wide range of sleep-related disorders, from difficulty falling asleep to difficulty sleeping through the night, insomnia, non-restorative sleep, or a significantly increased need for sleep. This could be related to a loss of signaling or increased OX signaling due to a disorder of the OX system.

Neurologically, the OX system is part of a finely tuned, complex network of different neurotransmitter systems. It therefore influences a large number of bodily functions. There is a known connection with many neurological diseases that are also associated with the consequences of a COVID-19 infection. It affects neurocognitive performance, and could thus be responsible for various symptoms that affect learning, memory and language, as well as influencing mood and emotions.

Modulation of OX levels could therefore have a therapeutic effect on these and other symptoms associated with PASC, and also mediate anti-inflammatory and immunomodulatory effects. Although there is little evidence of a link, and these considerations are so far only hypothetical, the overlap between PASC symptoms and symptoms caused by dysregulation of the OX system is so great that further investigation seems worthwhile. We hope that this work will lead to further research in this area in the future to confirm the hypothesis of a link.

## Figures and Tables

**Figure 1 biomedicines-13-00545-f001:**
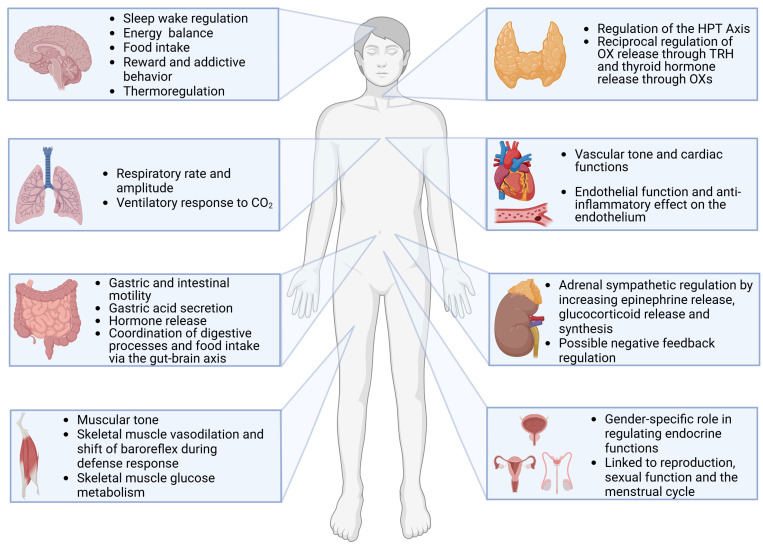
Occurrence of OXs and OXRs in the body, as well as function in the respective body area. It should be noted that many of the studies on the distribution and functions of OXs were carried out in animals, and that in humans these are therefore partly predicted functions and distributions based on comparisons and indications. Created in BioRender. Ruhrländer, J. (2025) https://BioRender.com/i52m727 (accessed on 14 December 2024).

**Figure 2 biomedicines-13-00545-f002:**
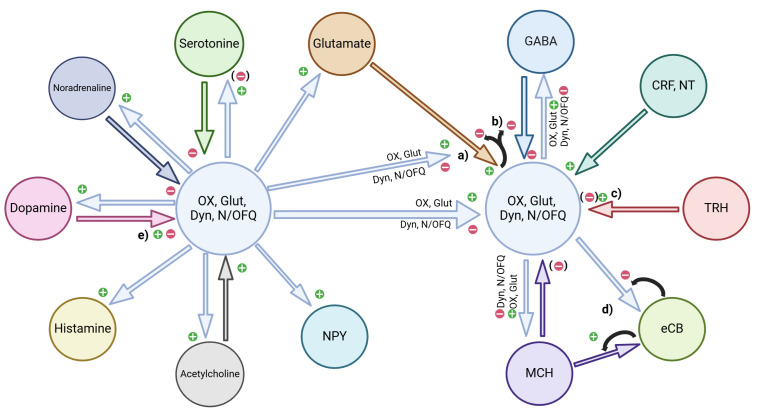
Schematic representation of the OX network with direct excitatory (+) and inhibitory (−) inputs and outputs and indirect inhibitory (−) inputs and outputs. The large coloured arrows show the direction of action of regular signal transmission. The black arrows symbolize a feedback effect of the released transmitter on the presynaptic neuron. (**a**) Synaptic modulation can indirectly regulate excitatory synaptic inputs, or (**b**) released Glut can inhibit Glut and GABA release by (auto)inhibition. (**c**) TRH has a direct excitatory effect on OX neurons but could also have an indirect inhibitory effect via an increase in the activity of GABAergic interneurons, depending on the site of release. (**d**) eCB release on downstream (OX) neurons has an inhibitory effect on OX neurons via retrograde transmission, while they have an excitatory effect on MCH neurons. (**e**) DA shows a dose-dependent bidirectional effect on OX neurons. Created in BioRender. Ruhrländer, J. (2025) https://BioRender.com/n12i290 (accessed on 14 December 2024).

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
