# Peer review of "The Orexin System and Its Impact on the Autonomic Nervous and Cardiometabolic System in Post-Acute Sequelae of COVID-19"

_biomedicines, 2025, doi:10.3390/biomedicines13030545_

Round 1

Reviewer 1 Report

Comments and Suggestions for Authors

The manuscript entitled The Orexin System and its Impact on the Autonomic Nervous and Cardiometabolic System in Post-Acute Sequelae of COVID-19  is presented for the peer review. The manuscipt is well wriiten and structured.  My suggestion is to add phrase about genetics of OXR regulation in humans including cell models.

Second suggestion dealing with cytokine- OX interaction during cytokine storm. 

My minor The Orexin System and its Impact on the Autonomic Nervous

2
and Cardiometabolic System in Post-Acute Sequelae of 3
COVID-

19The Orexin System and its Impact on the Autonomic Nervous

2
and Cardiometabolic System in Post-Acute Sequelae of 3
COVID-19

Author Response

Response to Reviewer 1 Comments

1. Summary

Thank you very much for taking the time to review this manuscript, for your comments and your work, we really appreciate it.

Please find the detailed responses below and the corresponding revisions/corrections highlighted/in track changes in the re-submitted files.

Comments 1: The manuscript entitled The Orexin System and its Impact on the Autonomic Nervous and Cardiometabolic System in Post-Acute Sequelae of COVID-19  is presented for the peer review. The manuscipt is well wriiten and structured.  My suggestion is to add phrase about genetics of OXR regulation in humans including cell models.

Response 1: Thank you for this comment. The genetic regulation of OXR is indeed an interesting aspect, unfortunately there are generally only a few models with human cells. Nevertheless, I have added a few text passages in the appropriate places. You can find them under Page 5, Lines 208-213 and Page 10, Lines 417-425

Comments 2: Second suggestion dealing with cytokine- OX interaction during cytokine storm. 

Response 2: This is also an important note, and we have gone into the cytokine-orexin interaction in more detail at the appropriate point in the text. You can find it under Page 9, Line 388 to Page 10, Line 412.

In addition, I have corrected three errors in the text, on page 1, lines 24-26 there was an error caused by shortening the text, where it said

“In 1998, a pair of neuropeptides was first described by two independent groups (1,2) which were detected in neurons of the dorsolateral hypothalamus. These peptides showed structural similarities to the peptide family of integrins referred to as hypocretins.”

And I have corrected it to:

In 1998, a pair of neuropeptides was first described by two independent groups (1,2). Due to their expression in neurons of the dorsolateral hypothalamus and structural similarity to the peptide family of integrins, they were initially referred to as hypocretins.

Then I removed an unnecessary duplication on Page 6, Lines 227-229:

An imbalance in the OX system can therefore lead to a variety of metabolic disorders such as dyslipidemia, showed by studies that linked OX gene expression with hypertriglyceridemia (46) and insulin resistance (47) and could be important for the development of type 2 diabetes (42), for which, like for insulin resistance and lipid dysregulation there is an increased risk following COVID-19 infection (48). Also, increased risk and higher burden of dyslipidemia have been associated with COVID-19 (49) as well as the impact of statin therapy on lipid metabolism and therefore COVID-19 disease (50).

I have removed the duplication in the new text, it now reads:

An imbalance in the OX system can therefore lead to a variety of metabolic disorders such as dyslipidemia, showed by studies that linked OX gene expression with hypertriglyceridemia (50) and insulin resistance (51) and could be important for the development of type 2 diabetes (46), for which, like for insulin resistance there is an increased risk following COVID-19 infection (52). Also, increased risk and higher burden of dyslipidemia have been associated with COVID-19 (53) as well as the impact of statin therapy on lipid metabolism and therefore COVID-19 disease (54).

Thirdly I added the word COVID in Line 358:

From: Not only acute COVID-19 infections but also vaccinations can trigger various symptoms […].

To: Not only acute COVID-19 infections but also COVID vaccines can trigger various symptoms […]

I made the following changes to the text on the suggestions of the two other reviewers:

I have adapted and expanded the discussion section once again in order to work out our hypothesis more clearly and to explain once again why we put it forward. At the very end, I emphasized once again that this hypothesis is purely speculative and that more research is needed to confirm it. You can find it from Page 10, Line 431 to Page 11, Line 480.

And I have added two more references on Page 9, Lines 365-370.

Reviewer 2 Report

Comments and Suggestions for Authors Review Manuscript ID: biomedicines-3452500     Comment tu the authors:

The manuscript titled: "The Orexin System and its Impact on the Autonomic Nervous and Cardiometabolic System in Post-Acute Sequelae of COVID-19" by Jana Ruhrländer et al. is well-written, with a clear and logical progression of information. My only suggestion pertains to the discussion section, where the authors could provide a more comprehensive formulation of their hypotheses following the five sessions that report observations from the literature. Enhancing this aspect would strengthen the manuscript and facilitate its publication.

Author Response

Response to Reviewer 2 Comments

1. Summary

Thank you very much for taking the time to review this manuscript, for your comments and your work, we really appreciate it.

Please find the detailed responses below and the corresponding revisions/corrections highlighted/in track changes in the re-submitted files.

Comments 1: The manuscript titled: "The Orexin System and its Impact on the Autonomic Nervous and Cardiometabolic System in Post-Acute Sequelae of COVID-19" by Jana Ruhrländer et al. is well-written, with a clear and logical progression of information. My only suggestion pertains to the discussion section, where the authors could provide a more comprehensive formulation of their hypotheses following the five sessions that report observations from the literature. Enhancing this aspect would strengthen the manuscript and facilitate its publication.

Response 1: Thank you for this suggestion. Indeed, a clearer formulation and justification of the hypothesis would give the manuscript more clarity. I have therefore clearly rewritten the discussion section to emphasise once again what led us to formulate the hypothesis and to reinforce it. You can find it from Page 10, Line 431 to Page 11, Line 480. 

In addition, I have corrected three errors in the text, on page 1, lines 24-26 there was an error caused by shortening the text, where it said

“In 1998, a pair of neuropeptides was first described by two independent groups (1,2) which were detected in neurons of the dorsolateral hypothalamus. These peptides showed structural similarities to the peptide family of integrins referred to as hypocretins.”

And I have corrected it to:

In 1998, a pair of neuropeptides was first described by two independent groups (1,2). Due to their expression in neurons of the dorsolateral hypothalamus and structural similarity to the peptide family of integrins, they were initially referred to as hypocretins.

Then I removed an unnecessary duplication on Page 6, Lines 227-229:

An imbalance in the OX system can therefore lead to a variety of metabolic disorders such as dyslipidemia, showed by studies that linked OX gene expression with hypertriglyceridemia (46) and insulin resistance (47) and could be important for the development of type 2 diabetes (42), for which, like for insulin resistance and lipid dysregulation there is an increased risk following COVID-19 infection (48). Also, increased risk and higher burden of dyslipidemia have been associated with COVID-19 (49) as well as the impact of statin therapy on lipid metabolism and therefore COVID-19 disease (50).

I have removed the duplication in the new text, it now reads:

An imbalance in the OX system can therefore lead to a variety of metabolic disorders such as dyslipidemia, showed by studies that linked OX gene expression with hypertriglyceridemia (50) and insulin resistance (51) and could be important for the development of type 2 diabetes (46), for which, like for insulin resistance there is an increased risk following COVID-19 infection (52). Also, increased risk and higher burden of dyslipidemia have been associated with COVID-19 (53) as well as the impact of statin therapy on lipid metabolism and therefore COVID-19 disease (54).

Thirdly I added the word COVID in Line 358:

From: Not only acute COVID-19 infections but also vaccinations can trigger various symptoms […].

To: Not only acute COVID-19 infections but also COVID vaccines can trigger various symptoms […]

I made the following changes to the text on the suggestions of the two other reviewers:

At the very end of the discussion, I emphasised once again that this hypothesis is purely speculative and that more research is needed to confirm it.

And I have added two more references on Page 9, Lines 365-370.

And I added two parts on genetic regulation of OXR in human with cell models, you can find it under: Page 5, Lines 208-213 and Page 10, Lines 417-425

I delved deeper into the topic of OX-cytokine interaction. You can find it under Page 9, Line 388 to Page 10, Line 412.

Reviewer 3 Report

Comments and Suggestions for Authors

In the present manuscript, Ruhrlander and co-authors have reviewed the complex pathophysiology of orexins and suggested that their dysregulation may be associated with the development of post-Covid-19 syndrome.

The review is well written and clear, and the references are appropriate and up to date. I have no major criticisms, only a few suggestions.

As there is little clinical evidence for a disruption of the orexin system in patients with post-Covid 19 syndrome, I would suggest that the authors emphasise that, although plausible and suggestive, their assumptions about a possible role played by orexins in maintaining the plethora of symptoms that characterise post-Covid are at present speculative and need validation. In addition, to support their hypothesis, I think it may be useful to also cite in their paper a very recent case-report of a girl diagnosed with narcolepsy (and with low CSF orexin A levels) following Covid-19 (PMID:39729629). 

Author Response

Response to Reviewer 3 Comments

1. Summary

Thank you very much for taking the time to review this manuscript, for your comments and your work, we really appreciate it.

Please find the detailed responses below and the corresponding revisions/corrections highlighted/in track changes in the re-submitted files.

Comments 1: As there is little clinical evidence for a disruption of the orexin system in patients with post-Covid 19 syndrome, I would suggest that the authors emphasise that, although plausible and suggestive, their assumptions about a possible role played by orexins in maintaining the plethora of symptoms that characterise post-Covid are at present speculative and need validation. In addition, to support their hypothesis, I think it may be useful to also cite in their paper a very recent case-report of a girl diagnosed with narcolepsy (and with low CSF orexin A levels) following Covid-19 (PMID:39729629). 

Response 1: Thank you for these suggestions. I have reworked the discussion section to make our hypothesis clearer. At the very end I have also made it clear that this is a purely speculative assumption and that further research will be needed in the future to prove these connections. You can find this part on page 11, lines 476-480.

I have also added the reference you suggested and another case report on narcolepsy following COVID infection, both of which were so recent that I was unaware of them as I had completed my literature search earlier. These are absolutely enriching for the manuscript and an important addition, so I would like to thank you very much for bringing them to my attention.

You can find it on Page 9, Lines 365-370. 

In addition, I have corrected three errors in the text, on page 1, lines 24-26 there was an error caused by shortening the text, where it said

“In 1998, a pair of neuropeptides was first described by two independent groups (1,2) which were detected in neurons of the dorsolateral hypothalamus. These peptides showed structural similarities to the peptide family of integrins referred to as hypocretins.”

And I have corrected it to:

In 1998, a pair of neuropeptides was first described by two independent groups (1,2). Due to their expression in neurons of the dorsolateral hypothalamus and structural similarity to the peptide family of integrins, they were initially referred to as hypocretins.

Then I removed an unnecessary duplication on Page 6, Lines 227-229:

An imbalance in the OX system can therefore lead to a variety of metabolic disorders such as dyslipidemia, showed by studies that linked OX gene expression with hypertriglyceridemia (46) and insulin resistance (47) and could be important for the development of type 2 diabetes (42), for which, like for insulin resistance and lipid dysregulation there is an increased risk following COVID-19 infection (48). Also, increased risk and higher burden of dyslipidemia have been associated with COVID-19 (49) as well as the impact of statin therapy on lipid metabolism and therefore COVID-19 disease (50).

I have removed the duplication in the new text, it now reads:

An imbalance in the OX system can therefore lead to a variety of metabolic disorders such as dyslipidemia, showed by studies that linked OX gene expression with hypertriglyceridemia (50) and insulin resistance (51) and could be important for the development of type 2 diabetes (46), for which, like for insulin resistance there is an increased risk following COVID-19 infection (52). Also, increased risk and higher burden of dyslipidemia have been associated with COVID-19 (53) as well as the impact of statin therapy on lipid metabolism and therefore COVID-19 disease (54).

Thirdly I added the word COVID in Line 358:

From: Not only acute COVID-19 infections but also vaccinations can trigger various symptoms […].

To: Not only acute COVID-19 infections but also COVID vaccines can trigger various symptoms […]

I made the following changes to the text on the suggestions of the two other reviewers:

I have adapted and expanded the discussion section once again in order to work out our hypothesis more clearly and to explain once again why we put it forward. You can find it from Page 10, Line 431 to Page 11, Line 480.

And I added two parts on genetic regulation of OXR in human with cell models, you can find it under: Page 5, Lines 208-213 and Page 10, Lines 417-425

I delved deeper into the topic of OX-cytokine interaction. You can find it under Page 9, Line 388 to Page 10, Line 412.